

# A large and unusually colored new snake species of the genus *Tantilla* (Squamata; Colubridae) from the Peruvian Andes

Claudia Koch[1] and Pablo J. Venegas[2]

[1] Department of Herpetology, Zoologisches Forschungsmuseum Alexander Koenig (ZFMK), Bonn, Germany
[2] Department of Herpetology, Centro de Ornitología y Biodiversidad (CORBIDI), Lima, Peru

## ABSTRACT

A new colubrid species of the genus *Tantilla* from the dry forest of the northern Peruvian Andes is described on the basis of two specimens, which exhibit a conspicuous sexual dimorphism. *Tantilla tjiasmantoi* sp. nov. represents the third species of the genus in Peru. The new species is easily distinguished from its congeners by the combination of scalation characteristics and the unusual transversely-banded color pattern on the dorsum. A detailed description of the skull morphology of the new species is given based on micro-computed tomography images. The habitat of this new species is gravely threatened due to human interventions. Conservation efforts are urgently needed in the inter-Andean valley of the Maranon River.

## INTRODUCTION

Seasonally dry tropical forests (SDTF) are characterised by a distinct seasonality with several months of arid-like conditions in which many plants lose their leaves (*Murphy & Lugo, 1986*). In South America, SDTFs are discontinuously distributed and can occupy large areas such as the Caatinga in northeastern Brazil or small fragments as being found in inter-Andean valleys of Peru or Ecuador (*Werneck et al., 2011*). Nevertheless, the different areas of South American SDTFs are very divers and the species compositions differ substantially (*Linares-Palomino, 2006*; *Miles et al., 2006*; *DRYFLOR, 2016*). However, SDTFs are considered as being one of the most threatened tropical ecosystems with a strong rate of annual deforestation (*Janzen, 1988*; *Miles et al., 2006*; *Pennington, Lewis & Ratter, 2006*; *Linares-Palomino, Oliveira-Filho & Pennington, 2011*; *DRYFLOR, 2016*). The Equatorial dry forest is one representative of this forest type, which expands from southern Ecuador to the northern part of Peru (*Brack, 1986*; *Särkinen et al., 2011*; *Venegas, 2005*), where it extends southward in two small stripes. One stripe continues along the coast west of the Andes, whereas the other penetrates the valley of the Marañón River and its tributaries. This North-to-South oriented valley is located in the Central Andes and bordered to the West and East by the Cordillera Occidental and the Cordillera Central, respectively. The inter-Andean dry forest expands from the Huancabamba Depression in

Corresponding author
Claudia Koch, c.koch@zfmk.de

the North along the flanks of the Chinchipe, Chamaya, Huancabamba and Utcubamba rivers and tributaries (Departments of Piura, Cajamarca, Amazonas) southward along the deep and narrow valley of the Marañón River to the Department of La Libertad. Although it has so far rarely been studied with respect to its flora and fauna, recent studies indicate that especially this inter-Andean part of dry forest, is home to a high proportion of endemic species (*Stattersfield et al., 1998*; *Bridgewater et al., 2003*; *Linares-Palomino, 2006*; *Koch, Venegas & Böhme, 2006*; *Koch, Venegas & Böhme, 2015*; *Koch et al., 2011*; *Koch et al., 2013*; *Venegas et al., 2008*; *Särkinen et al., 2011*). According to *Linares-Palomino (2006)* this region is home to 184 woody plant species, including 69 Peruvian endemics. Characteristic representatives of this xeric vegetation are drought-resistant trees (e.g., *Acacia, Ceiba, Cordia, Eriotheca, Hura, Prosopis*), cacti plants (e.g., *Armatocereus, Browningia, Espostoa*), dense shrubs (e.g., *Mimosa, Croton*) and ground vegetation layer (e.g., *Opuntia,* Poaceae). Unfortunately, this region is facing multiple and complex threats due to logging, agriculture and narcotics plantations, mining activities, and above all due to several planned hydroelectric projects that will lead to flooding of vast portions of the inter-Andean dry forest (*Q & V Ingenieros SAC, 2007*; *Finer & Jenkins, 2012*; *Lees et al., 2016*). These threats, together with the enormous lack of knowledge that still exists about its biodiversity, and the fact that no protected area has so far been designated in the inter-Andean part of the Equatorial dry forest (*IUCN & UNEP-WCMC, 2016*), highlights the urgent priority for conservation and research activities in this area.

To contribute to the knowledge of the peculiar herpetofauna of the Andean dry forest, we conducted fieldwork in inter-Andean valleys of the northern Peruvian Departments of Amazonas, Cajamarca, and La Libertad and surveyed 28 localities along a stretch of more than 350 km of the Marañón River and its tributaries. In two of the southernmost investigation areas in the Department of La Libertad, we collected each a specimen of a colubrid species, which was difficult to assign to any of the known genera based on traits of external morphology alone. However, a phylogenetic analysis based on 12S and 16S rRNA together with a comparison of the skull morphology via micro-computed tomography (micro-CT) scans revealed sufficient evidence to place it in the genus *Tantilla* Baird & Girard, 1853.

Currently, 61 species within the genus *Tantilla* are recognized (*Wilson & Mata-Silva, 2015*; *Mata-Silva & Wilson, 2016*). Twelve species occur in mainland South America, of which only two are found in Peru: *T. capistrata* Cope, 1875 and *T. melanocephala* (Linnaeus, 1758). Most species of the genus have either a uniformly colored or a longitudinally striped dorsal color pattern. Only *T. shawi* Taylor, 1949 from Mexico, *T. semicincta* (Duméril, Bibron & Duméril, 1854) from Colombia and Venezuela, and *T. supracincta* (Peters, 1863) from Colombia, Costa Rica, Ecuador, Nicaragua, and Panama have a transversely-banded color pattern on the dorsal part of the body (*Wilson, 1976*; *Wilson & Mata-Silva, 2015*).

Reviewing published information on morphological characteristics of all other species of the genus *Tantilla* (*Wilson, 1987*; *Wilson, 1999*; *Sawaya & Sazima, 2003*; *Townsend et al., 2013*; *Wilson & Mata-Silva, 2014*; *Wilson & Mata-Silva, 2015*) revealed that the collected specimens can be easily differentiated from others of the genus by its comparatively high number of ventral scales and other scalation characteristics, the relatively large size, and

transversely-banded color pattern. Herein a detailed description of the new species is given.

## MATERIALS AND METHODS

### Fieldwork, data sampling and Zoobank registration

The new species was detected during visual encounter surveys (*Crump & Scott, 1994*) during our fieldwork conducted between March 2009 and November 2010. Altitudes above sea level and geographic coordinates were recorded with a GPS (Garmin GPSMap 60CSx) using the geodetic datum WGS84. Air temperatures and humidity were taken with a digital thermo-hygrometer (Extech) using an external sensor. All the necessary research and collecting (0020-2009-AG-DGFFS-DGEFFS, 0424-2010-AG-DGFFS-DGEFFS) and export permits (003983-AG-DGFFS) for this study were issued by the Ministerio de Agricultura of the government of Peru (Ministerio de Agricultura). Both specimens were collected by hand, photographed in live and subsequently euthanized with the narcotic T61®. Tissue samples were extracted from the muscle of the lateral body and stored in 96% ethanol. Specimens were placed in 10% formalin for fixation for about 12 h, and ultimately stored in 70% ethanol. Finally the holotype (CORBIDI 7726) was deposited in the herpetological collection of the Centro de Ornitología y Biodiversidad, Lima, Peru (CORBIDI) and the paratype (ZFMK 95238) was deposited in the collection of the Zoologisches Forschungsmuseum Alexander Koenig, Bonn, Germany (ZFMK). The electronic version of this article in Portable Document Format (PDF) will represent a published work according to the International Commission on Zoological Nomenclature (ICZN), and hence the new names contained in the electronic version are effectively published under that Code from the electronic edition alone. This published work and the nomenclatural acts it contains have been registered in ZooBank, the online registration system for the ICZN. The ZooBank LSIDs (Life Science Identifiers) can be resolved and the associated information viewed through any standard web browser by appending the LSID to the prefix http://zoobank.org/. The LSIDs for this publication is: urn:lsid:zoobank.org:pub:00EBF842-3AFA-4913-B381-95BDF86DAFAB. The online version of this work is archived and available from the following digital repositories: PeerJ, PubMed Central and CLOCKSS.

Data on morphological traits of other South American colubrid species and genera were taken from *Peracca (1904)*, *Van Denburgh (1912)*, *Werner (1924)*, *Stuart & Bailey (1941)*, *Slevin (1942)*, *Smith & Taylor (1945)*, *Stuart (1949)*, *Stuart (1954)*, *Peters (1960)*, *Fritts & Smith (1969)*, *Peters, Donoso-Barros & Orejas-Miranda (1970)*, *Villa (1971)*, *Villa (1990)*, *Myers (1973)*, *Wilson (1976)*, *Wilson (1987)*, *Wilson (1999)*, *Wilson & Mena (1980)*, *Savage & Donnelly, 1988*, *Pérez-Santos & Moreno, 1988*, *Savage & Crother (1989)*, *McCranie & Wilson (1991a)*, *McCranie & Wilson (1991b)*, *McCranie & Wilson (1992)*, *McCranie & Villa (1993)*, *Cei (1993)*, *Ferrarezzi (1993)*, *Puorto & Ferrarezzi, 1993*, *Myers & Cadle (1994)*, *Smith & Campbell (1994)*, *Campbell, Camarillo & Ustach (1995)*, *Zaher (1996)*, *Zaher (1999)*, *Franco, Marques & Puorto (1997)*, *Campbell (1998)*, *Campbell & Smith (1998)*, *Starace (1998)*, *Kornacker (1999)*, *Savage (2002)*, *Sawaya & Sazima (2003)*,

*Köhler (2003)*, *Köhler (2008)*, *McCranie & Castaneda (2004)*, *Solorzano, 2004*, *Stafford (2004)*, *Lema, D'Agostini & Cappelari (2005)*, *Scott et al. (2006)*, *Cacciali, Carreira & Scott (2007)*, *Harvey et al. (2008)*, *Jansen & Köhler (2008)*, *Zaher et al. (2009)*, *Myers (2011)*, *Passos, Ramos & Pereira (2012)*, *Moura Ribeiro, Caldeira Costa & Magalhães Pirani (2013)*, *Townsend et al. (2013) Myers & McDowell (2014)*, *Wilson & Mata-Silva (2015)* and *Wilson & Mata-Silva (2014)*.

## Morphological analyses

Measurements of head and scales were taken with a digital caliper and rounded to the nearest 0.1 mm, snout–vent length and tail length were taken with a measuring tape and rounded to the nearest 1 mm. Morphometric and meristic characters are abbreviated as follows: SVL (snout–vent length, from tip of snout to cloaca); TL (tail length); HW (head width across supraoculars); HH (head height at highest part of head); HL (head length); DSN (distance from tip of snout to nostril); DNE (distance from nostril to anterior margin of eye); ED (eye diameter); MBD (body diameter at midbody); MTD (midtail diameter). The number of ventral scales was counted in longitudinal row from mental to anal plate, and the number of subcaudal scales was counted in longitudinal row from the cloaca to the tip of the tail (*Dowling, 1951*). The number of dorsal scales rows around the body was counted at three different points: (1) at a head's length behind the head; (2) at midbody; (3) at a head's length before the cloaca. A dissecting microscope was used to count and characterize small scales and to identify the number of teeth in the male paratype.

For obtaining information on skeletal morphology, specimens were X-rayed in 2D (Faxitron X-ray LX60) and in 3D by use of a micro-CT scanner (Bruker Skyscan 1272). Terminology for the skull structures was adopted from *Bullock & Tanner (1966)* and *Cundall & Irish (2008)*. The structures of the skull are abbreviated as follows: PMX (premaxilla); NA (nasal); SMX (septomaxilla); F (frontal); PFR (prefrontal); P (parietal); PO (postorbital); PRO (prootic); ST (supratemporal); SO (supraoccipital); EXO (exoccipital); Q (quadrate); MX (maxilla); ECP (ectopterygoid); PAL (palatine); MP (maxillary process of palatine); CHP (chanal process of palatine); PT (pterygoid); BS (basisphenoid); BO (basioccipital).

The partially everted hemipenes of the male paratype were removed from the specimen and prepared following *Zaher & Prudente (2003)*. Finally the organs were scanned with the micro-CT scanner. Left hemipenis was scanned dry, whereas right hemipenis was scanned in alcohol. Terminology for hemipenes morphology follows *Zaher (1999)*.

## Phylogenetic analysis

Genomic DNA was extracted from the collected tissue samples at the Center for Molecular Biodiversity Research of the ZFMK, using the DNeasy Blood & Tissue Kit (Qiagen) following the standardized extraction protocol provided by the manufacturer. Two mitochondrial markers, 12S rRNA and 16S rRNA, and the nuclear gene RAG1 (recombination-activating gene 1) were amplified using polymerase chain reactions (PCR) having a final volume of 20 μl and carried out either on a GeneAmp 2,700 or on a Biometra thermal cycler. The QIAGEN® Multiplex PCR Kit (HotStarTaq® DNA Polymerase,

Multiplex PCR Buffer with 6 mM MgCl$_2$, dNTP Mix, 2 µl Q-solution, 2.3 µl ultrapure H$_2$O) was used for the reaction for all three genes with 1.6 µl of each primer, and 2.5 µl of extracted DNA. To amplify the 12S fragment the roughly universal primer pair 12SA-L (light chain) and 12SB-H (heavy chain) of *Kocher et al. (1989)* was used. The 16S fragment was amplified using the likewise universal primer pair 16sar-L (light chain) and 16sbr-H (heavy chain) of *Palumbi et al. (1991)*. For amplification of RAG1 the primer pair RAG1f2 (light chain) and RAG1r3 (heavy chain) of *Schulte & Cartwright (2009)* were used. Amplification with the 12S primer pair started with an initial denaturation step for 90 s at 94 °C, and 38 cycles were run with denaturing for 45 s at 94 °C, annealing for 60 s at 50 °C, elongation for 120 s at 74 °C, the final elongation for 300 s at 74 °C, and cooling at 10 °C. Amplification with the 16S primer pair started with an initial denaturation step for 900 s at 95 °C, followed by 15 cycles of denaturation for 35 s at 94 °C, annealing for 90 s at 60 °C, elongation for 90 s at 72 °C, plus 25 cycles of denaturation for 35 s at 94 °C, annealing for 90 s at 45 °C, elongation for 90 s at 72 °C, the final elongation for 600 s at 72 °C, and cooling at 10 °C. Amplification with the RAG1 primer pair started with an initial denaturation step for 900 s at 95 °C, and 40 cycles were run with denaturing for 20 s at 94 °C, annealing for 50 s at 60 °C, elongation for 90 s at 72 °C, the final elongation for 600 s at 72 °C, and cooling at 10 °C. After the PCR, each sample proving successful DNA amplification in an agarose gel electrophoresis, was purified for sequencing using the QIAquick PCR Purification Kit (Qiagen). Subsequently, the samples were sequenced by Macrogen Europe Laboratory (Amsterdam, Netherlands). Obtained sequences were checked with the original chromatograph data using BioEdit 7.5.2 (*Hall, 1999*). The 12S rRNA and 16S rRNA data was supplemented with sequences of 48 species representing 27 genera of American colubrid snakes obtained from GenBank. Accession numbers are provided in Table 1. Sequence alignment was done with MAFFT (*Katoh, Asimenos & Toh, 2009*) and manually corrected where necessary. Ambiguous sites (namely in the hypervariable regions of the 16S rRNA) were identified with Gblocks (*Castresana, 2000*) and excluded from the alignment during subsequent analyses, resulting in 350 bp and 438 bp for 12S and 16S, respectively. For each gene, GTR+I+G was chosen as model of nucleotide substitution by the Akaike information criterion using Modeltest (*Posada & Crandall, 1998*) as implemented in the package 'phangorn' for Cran R. Phylogenetic trees were inferred using MrBayes 3.2.6 (*Ronquist et al., 2012*), estimating model parameters separately for each gene by partitioning the data set. We used a random starting tree and four independent runs with a maximum of 10 million generations each, sampled every 1000. Runs were stopped when the average standard deviation of split frequencies had reached 0.01. Convergence of the Markov chains and effective sample sizes were checked with Tracer v1.6 (*Rambaut et al., 2014*) and the initial 25% of each run were discarded prior to building a consensus tree. In addition to the Bayesian inference (BI), phylogenies were also calculated with Maximum Likelihood (ML) via the RAxML BlackBox (*Stamatakis, Hoover & Rougemont, 2008*) using the partitioned data, the Gamma model of rate heterogeneity, and 100 bootstraps.

**Table 1** Taxa used for phylogenetic analysis and respective GenBank accession numbers.

| Species | 12S rRNA | 16S rRNA | RAG1 |
|---|---|---|---|
| CORBIDI 7726 | KY006875 | KY006877 | KY006874 |
| ZFMK 95238 | | KY006876 | KY006873 |
| *Chironius bicarinatus* | HM565744 | HM582206 | |
| *Chironius carinatus* | HM565745 | HM582207 | |
| *Chironius exoletus* | HM565746 | HM582208 | |
| *Chironius flavolineatus* | HM565747 | HM582209 | |
| *Chironius foveatus* | HM565748 | HM582210 | |
| *Chironius fuscus* | HM565749 | HM582211 | |
| *Chironius laevicollis* | HM565751 | HM582213 | |
| *Chironius monticola* | HM565753 | HM582214 | |
| *Chironius quadricarinatus* | HM565755 | HM582215 | |
| *Chironius scurrulus* | HM565756 | HM582216 | |
| *Clelia bicolor* | GQ457787 | GQ457729 | |
| *Coluber constrictor* | AY122667 | | |
| *Coluber flagellum* | AY122668 | | |
| *Coluber taeniatus* | AY122669 | | |
| *Dendrophidion percarinatus* | HM565757 | HM582217 | |
| *Drymarchon corais* | HM565758 | HM582218 | |
| *Drymobius rhombifer* | HM565761 | HM582220 | |
| *Drymoluber brazili* | HM565760 | HM582219 | |
| *Drymoluber dichrous* | HM565759 | HM582221 | |
| *Erythrolamprus aesculapii* | GQ457795 | GQ457736 | |
| *Gyalopion canum* | KR814624 | KR814641 | |
| *Lampropeltis getula* | AY122821 | | |
| *Lampropeltis mexicana* | FJ623962 | | |
| *Lampropeltis triangulum* | FJ623963 | | |
| *Leptodrymus pulcherrimus* | KR814627 | KR814649 | |
| *Leptophis ahaetulla* | HM565762 | HM582222 | |
| *Liophis elegantissimus* | GQ457808 | GQ457748 | |
| *Liophis jaegeri* | GQ457809 | GQ457749 | |
| *Mastigodryas bifossatus* | HM565763 | HM582223 | |
| *Mastigodryas boddaerti* | HM565764 | HM582224 | |
| *Mastigodryas dorsalis* | KR814625 | KR814650 | |
| *Oxybelis aeneus* | HM565765 | HM582225 | |
| *Oxybelis wilsoni* | KR814626 | KR814647 | |
| *Pituophis catenifer* | KU833245 | KU833245 | |
| *Pituophis lineaticollis* | AF512746 | AF512746 | |
| *Pseustes poecilonotus* | | KR815895 | |
| *Rhinobothryum lentiginosum* | HM565767 | HM582227 | |
| *Salvadora grahamiae* | AY122847 | | |
| *Scolecophis atrocinctus* | KR814619 | KR814642 | |

**Table 1** (*continued*)

| Species | 12S rRNA | 16S rRNA | RAG1 |
|---|---|---|---|
| *Senticolis triaspis* | AY122848 | | |
| *Spilotes pullatus* | HM565768 | HM582228 | |
| *Stenorrhina freminvillei* | HM565769 | | |
| *Symphimus leucostomus* | KR814618 | KR814651 | |
| *Tantilla armillata* | KR814613 | KR814644 | |
| *Tantilla impensa* | KR814614 | KR814645 | |
| *Tantilla melanocephala* | AF158424 | AF158491 | |
| *Tantilla vermiformis* | KR814615 | KR814646 | |
| *Xenodon neuwiedi* | GQ457841 | GQ457779 | |

**Table 2** Scale counts and measurements of the type specimens of *Tantilla tjiasmantoi* sp. nov.

| | ♀ Holotype CORBIDI 7726 | ♂ Paratype ZFMK 95238 |
|---|---|---|
| Dorsal scale rows | 15-15-15 | 15-15-15 |
| Ventrals | 182 | 179 |
| Subcaudals | 57 | 65 |
| Supralabials | 7 | 7 |
| Infralabials | 7 | 6-7 |
| Internasals | 2 | 2 |
| Prefrontals | 2 | 2 |
| Preoculars | 1 | 1 |
| Postoculars | 2 | 2 |
| Supraoculars | 1 | 1 |
| Suboculars | Absent | Absent |
| Loreal | Absent | Absent |
| Anterior temporals | 1 | 1 |
| Posterior temporals | 1 | 1 |
| Sublinguals (paired) | 2 | 2 |
| SVL | 513 mm | 198 mm |
| TL | 125 mm | 56 mm |
| TL/Total length | 0.2 | 0.22 |
| HW | 11.9 mm | 4.7 mm |
| HL | 16.3 mm | 8.1 mm |
| HW/HL | 0.73 | 0.58 |

## RESULTS

### Morphological analyses

As typical for the genus *Tantilla*, the two specimens possess a number of 15 smooth dorsal scale rows throughout the body, one preocular, no loreal, no suboculars, 1+1 temporals, a divided cloacal shield, paired subcaudals (Table 2). Additionally, the skull of the new species is composed of similar bones and bone structures as other species of the genus *Tantilla*. A comparison with three other congeners (*T. capistrata, T. melanocephala* and *T. relicta*) reveals great similarity to our new species with only minor differences in the

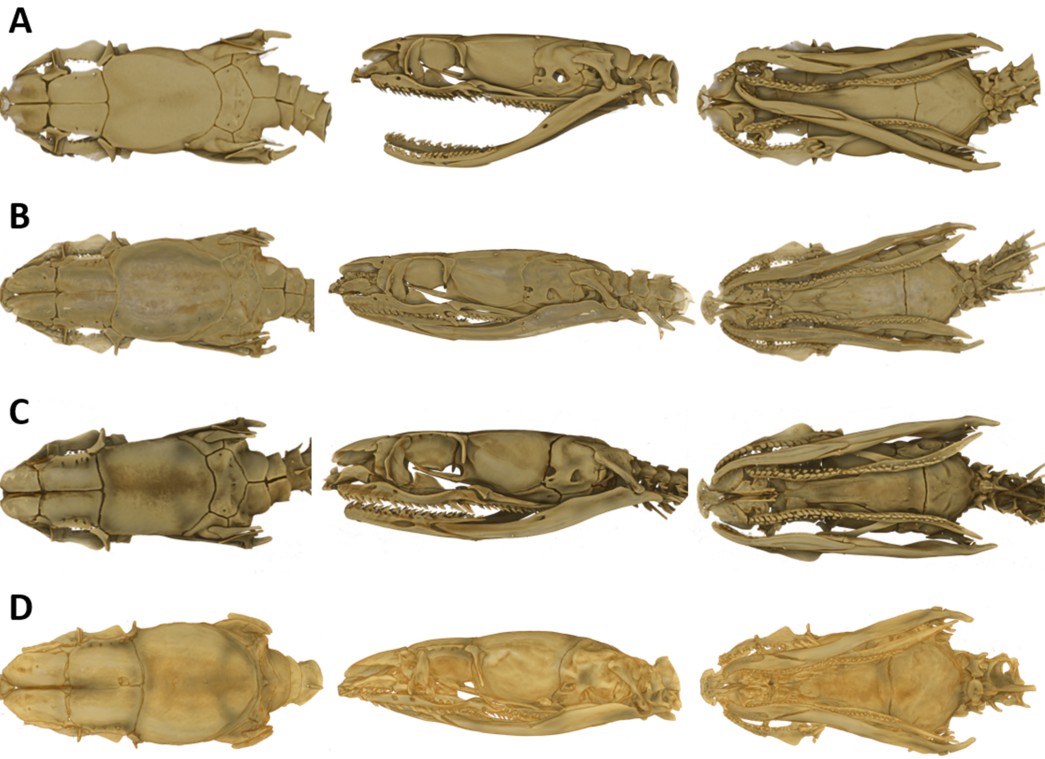

**Figure 1** Micro-CT images of the dorsal (left), lateral (middle), and ventral (right) views of the skull of four species of *Tantilla: T. tjiasmantoi* sp. nov. from Peru (ZFMK 95238, A), *T. capistrata* from Lambayeque, Peru (ZFMK 85028, B), *T. melanocephala* from Santa Cruz, Bolivia (ZFMK 75041, C) and *T. relicta* from Florida, USA (ZFMK 84387, D).

shape or size of some bones (Fig. 1). Scalation characteristics and dorsal color pattern are very similar in both specimens. Nevertheless, they show conspicuous differences in body size and ventral coloration which are most likely due to sexual dimorphism and/or age difference.

## Phylogenetic analysis

Fragments of the mitochondrial gene 16S (526 bps) and nuclear gene RAG1 (1,050 bps) of both specimens were compared. 16S showed no differences between specimens, and RAG1 revealed only a single base pair variation. The strong genetic similarity coupled with the weak morphological variation suggests these specimens are the same species.

The Bayesian consensus tree (Fig. 2) obtained from 788 bp of mitochondrial DNA (12S and 16S rRNA) was based on 3,600 sampled trees and effective sample sizes were >1,000, which indicated good mixing of the Markov chains. Although the topologies obtained from BI and ML (Fig. S1) differ and the nodes were generally not very well resolved in terms of posterior probabilities and bootstrap values, the four species of *Tantilla* formed a well-supported clade including our two specimens (CORBIDI 7726 and ZFMK 95238). However, node support is lacking between the different species of *Tantilla*, obscuring the exact relationship between these taxa.

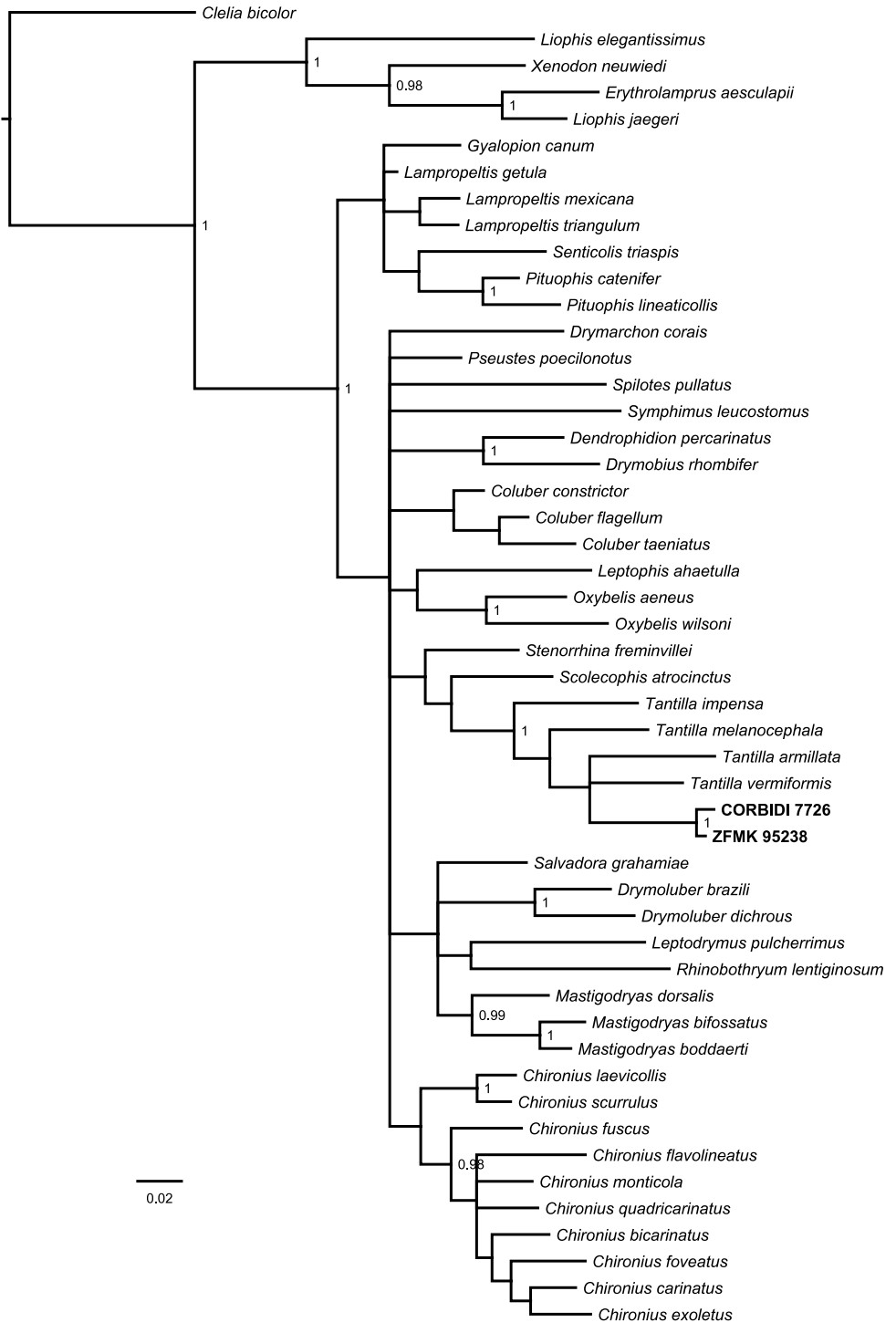

**Figure 2** **Bayesian consensus tree based on 788 bp of mitochondrial DNA (12S and 16S rRNA) of our specimens (CORBIDI 7726 and ZFMK 95238) and 48 further species representing 27 genera of American colubrid snakes.** Numbers at nodes are the Bayesian posterior probabilities (values <0.95 not shown).

### *Tantilla tjiasmantoi sp. nov.*

urn:lsid:zoobank.org:act:15B70DA0-55D5-47D9-8383-453869F3B530

### *Holotype*

CORBIDI 7726 from Pías, Province Pataz, Department of La Libertad, Peru (07°53′56.6″S, 77°34′43.8″W, 1,726 m a.s.l), collected by E. Hoyos Granda, A. Beraún and C. Koch on 15 January 2010.

### *Paratype*

ZFMK 95238, from Santa Rosa/El Tingo (Marcamachay), Province Cajabamba, Department of La Libertad, Peru (07°21′59.3″S, 77°53′53.0″W, 1,154 m a.s.l.), collected by M. Palacios Panta and C. Koch on 13 October 2010.

### *Diagnosis and definition*

This comparatively large *Tantilla* is distinguished from its congeners by the following combination of characters: (1) maximum known SVL of 513 mm and total length of 638 mm; (2) 179–182 ventrals; (3) 57–65 paired subcaudals; (4) 7 supralabials; (5) eyes small, not visible from below, with round pupils; (6) dorsals smooth, without keels or apical pits, rhomboid, in 15 rows throughout the body; (7) loreals absent; (8) suboculars absent; (9) 2 postoculars; (10) 1 + 1 temporals; (11) cloacal plate divided; (12) hypapophyses absent on posterior vertebrae; (13) hemipenes single; (14) body with dark bands that are not closed on ventral side; (15) conspicuous sexual dimorphism present.

With a maximum known total length of 638 mm *Tantilla tjiasmantoi* sp. nov. is among the largest species in the genus, only *T. shawi*, *T. impensa,* and *T. semicincta* reach similar or even larger total length. It can further be easily distinguished from all congeners except for *T. shawi, T. semicincta*, and *T. supracincta* by having a transversely-banded color pattern on the dorsal part of the body. The higher number of subcaudals (57–65) differentiates the new species from *T. shawi* (48–50) and the higher number of ventrals (179–182) further distinguishes it from *T. impensa* (162–172), *T. semicincta* (161–176) and *T. supracincta* (138–151), as well as from the Peruvian species *T. capistrata* (130–156) and *T. melancocephala* (125–177). From the other species occurring in mainland South America, it can further be distinguished by a higher number of ventrals as compared to *T. alticola* (128–145), *T. andinista* (157), *T. boipiranga* (156–167), *T. miyatai* (165–172), *T. nigra* (137), *T. petersi* (172), *T. insulamontana* (144–157), *T. reticulata* (158–173).

### *Description of holotype*

An adult female with a SVL of 513 mm; TL 125 mm; HL 16.3 mm; HW 12.2 mm; HH 6.1 mm; TL/Total Length 0.2; SVL/HL 31.5; SVL/HW 42.1; SVL/HH 84.1; HW/HL 0.75; HH/HL 2.7; ED 1.5 mm; HL/ED 10.9; HW/ED 8.1; MBD 11.5 mm; SVL/MBD 44.6; DSN 1.5 mm; DNE 3 mm; MTD 2.9 mm.

Body robust, tail long, body and tail round in cross-section; dorsal scales in 15-15-15-rows, without reduction, rhomboid, smooth, lacking keels or apical pits; 182 ventrals; tail distinctly smaller in diameter than the body, long and tapering, tail spine pointed; 57 paired subcaudals; cloacal plate divided.

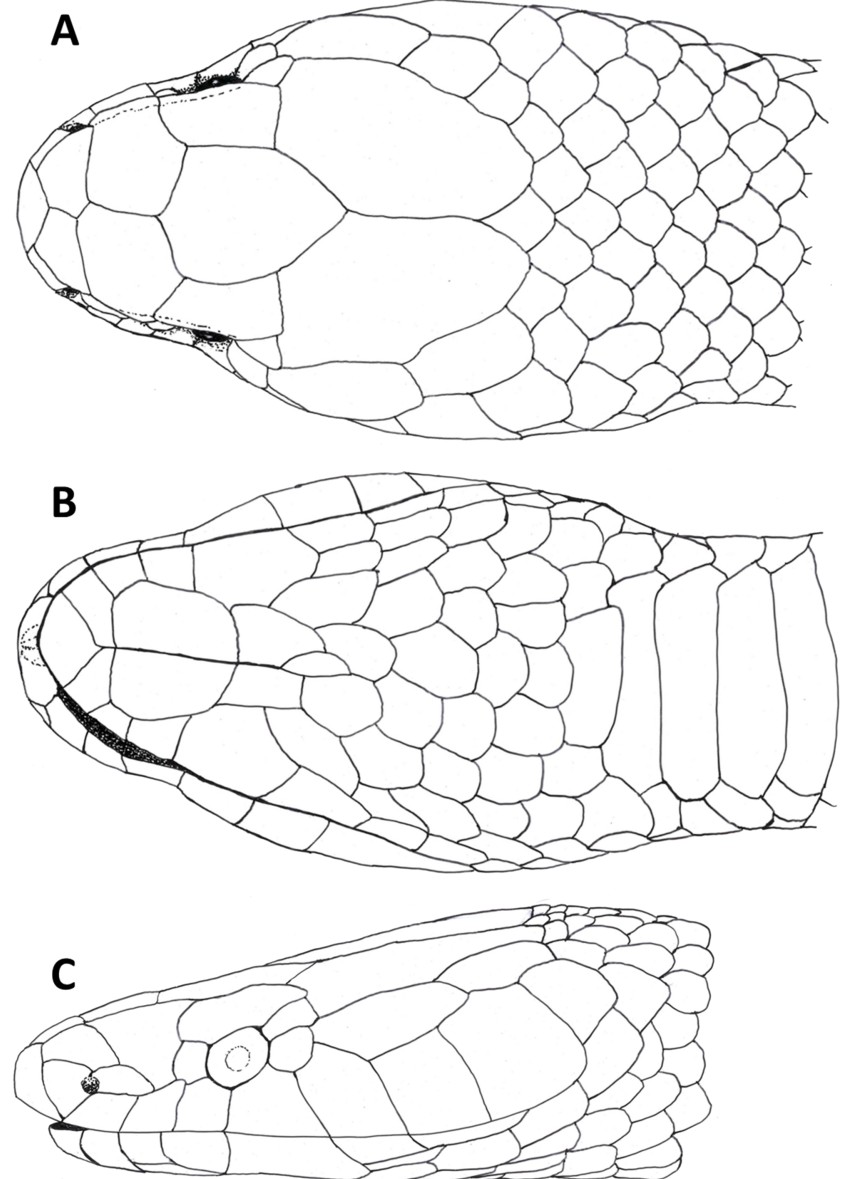

**Figure 3** Dorsal (A), ventral (B) and lateral (C) views of head of female holotype of *Tantilla tjiasmantoi* sp. nov. from La Libertad, Peru (CORBIDI 7726).

Head (Fig. 3) distinct from the body and depressed, laterally broadened behind the eyes. Rostral visible in dorsal view, slightly triangular with dorsal termination subacute, about 1.9 times wider than high. Snout rounded in dorsal view. Two internasals, subequal, about twice as broad as long, laterally in contact with anterior and posterior nasals. Prefrontals large, 1.2 times broader than long, laterally contacting postnasal, second supralabial and preocular. Frontal slightly hexagonal with dorsal termination acute, about 1.2 times longer than wide, laterally in contact with supraoculars, laterodorsally in contact with parietals. Supraoculars about 2.1 times wider than high. Parietals large, 1.6 times longer than wide, median suture about as long as frontal length. Nostrils located in posterior

portion of anterior nasals. Anterior nasals about as high as broad, contacting rostral, internasal, postnasal, and first supralabial. Posterior nasals about 1.8 times longer than high, contacting anterior nasal, internasal, prefrontal, and first and second supralabial. No loreal. Eyes small with round pupils, partly visible in dorsal view, surrounded by one preocular, one supraocular, two postoculars, and third and fourth supralabial. No suboculars. Preoculars almost squarish. Superior postoculars, about 1.5 times longer than high; lower postoculars slightly pentagonal to almost circular. Supralabials 7/7, increasing in size posteriard, last one very large, about three times higher and 2.2 times broader than first supralabial, second contacting postnasal and preocular, third contacting prefrontal and eye, fourth contacting eye and lower postocular, fifth contacting lower postocular and anterior temporal, sixth contacting anterior temporal, seventh contacting anterior and posterior temporals, and first scale of dorsolateral and lateral body scale row, respectively. 1 + 1 temporals, almost rectangular, separating supralabials 5–7 from parietals, anterior temporal about 2.5 times longer than high on both sides, 1.3 times longer than posterior one on left side and almost twice as long as posterior one on right side of head, posterior temporal 1.8 times longer than high on left side and about 1.1 times longer than high on right side of head. Mental subrhombical, 1.4 times broader than long, separated from chinshields by first pair of infralabials, which contact each other along the ventral midline. Infralabials 7/7, fourth largest, second smallest, first to fourth contacting anterior pair of chinshields; fourth infralabial contacting posterior chinshield and first gular scale. Two pairs of almost rectangular chinshields; anterior chinshields about 1.7 times longer than wide and 1.3 times longer than posterior chinshields; posterior chinshields about 2.2 times longer than wide, laterally contacting fourth infralabial, dorsally separated from the ventrals by four gular scales.

Trunk vertebrae 185; hypapophyses present on anterior 1/5 of trunk vertebrae, absent on posterior vertebrae; caudal vertebrae 56 (Fig. 4).

*Coloration*

In live, the dorsal ground color of head, body and tail is orange-yellowish, slightly paler laterally, most scales on body and tail with reddish-brown outlines; there are about 27 blackish dorsal crossbars on the body that are four to seven scales in length and are stretched across all dorsal scale rows except the most lateral row, slightly longer than ground color interspaces, fused in some parts of the body to form a zigzag band, slightly mottled in some parts with yellow. There are 12 dark tail blotches, reaching to subcaudals on both sides, fused in median part of the tail along the midline to form a zigzag band. Head with a large dark dorsal t-shirt-shaped blotch covering frontal, supraoculars, most of parietals except for the most posterior parts, and posterior part of prefrontals, the dark blotch is laterally extended at eye level, covering orbit, preocular, third supralabial, and adjacent parts of second and fourth supralabials, respectively. Infralabials, rostral and mental yellowish, except for blackish region surrounding the lingual groove. The ventral scales of head and body and subcaudal scales are cream-colored with dark dotted outlines in some parts. The coloration of the tongue is black to grayish-black (Fig. 5).

 

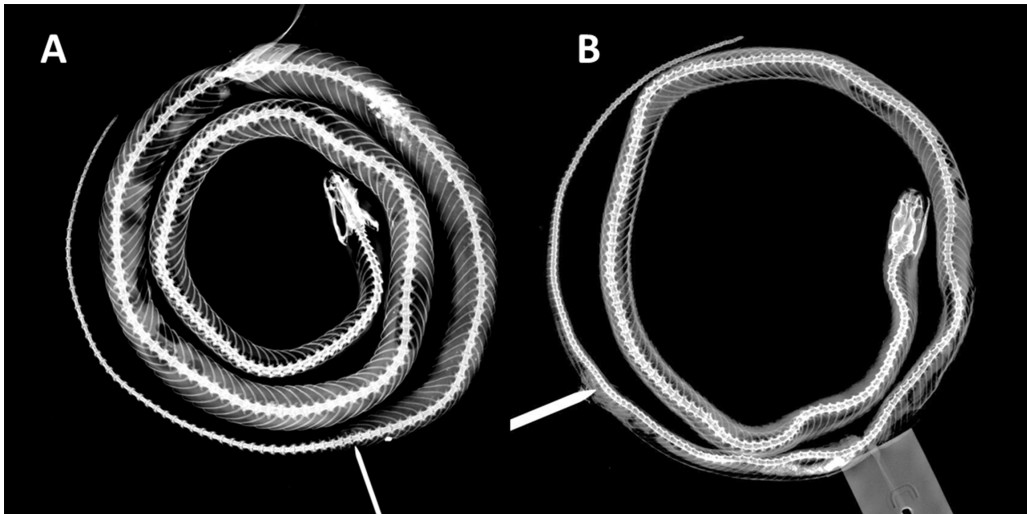

**Figure 4** X-ray photographies of body of *Tantilla tjiasmantoi* sp. nov. from La Libertad, Peru (needle marks the cloaca): female holotype CORBIDI 7726 (A) and male paratype ZFMK 95238 (B).

In preservative, the dorsal pattern on body and tail of a light ground color with dark crossbands remains unchanged, likewise the dark coloration of the head; the orange-yellowish dorsal ground color changed to cream and the darker outlines of most dorsal scales disappeared; the ventral coloration changed to grayish-white in some parts (Fig. 5). The coloration of the tongue changed to gray.

### Variation

The single paratype is a small male with a conspicuous sexual dimorphism in body size and ventral coloration compared to the holotype. As the hemipenes ornamentation is as detailed as in other adult specimens of the genus *Tantilla* we assumed it to be already sexually mature. Intraspecific variation in scale counts and measurements is shown in Table 2. The paratype further differs from the holotype in the following characters: SVL/HL 24.4; HW/HL 0.58; ED 1.1 mm; HL/ED 7.4; HW/ED 4.3; MBD 3.5 mm; SVL/MBD 56.6; DSN 1.0 mm; DNE 1.3 mm; MTD 1.3 mm. On the right side this specimen has only 6 infralabials, of which the third one is the largest.

The dorsal ground coloration of the paratype (Fig. 5) in live is bright yellow and there are 35 blackish dorsal crossbars on the body and 16 dark tail blotches, some of the latter are fused to form a zigzag band, the black parts are usually not mottled with yellow. The t-shirt-shaped dark pattern on the dorsal surface of the head is also present in this specimen, but there is a dark vertical line on the left side of the head covering whole sixth and adjacent parts of seventh supralabial. On left side of the head posterior part of second infralabial and infralabials three to five are dark, on right side of head the posterior part of third infralabial and the infralabials four to six are dark. The ventral coloration differs from the holotype as the dark dorsal coloration of the paratype continues on the ventral scales, and thus sometimes forming complete rings around the body or creating a checkerboard pattern on the ventral surface of black and cream squares. There are 179 trunk vertebrae and 70 caudal vertebrae (Fig. 4).

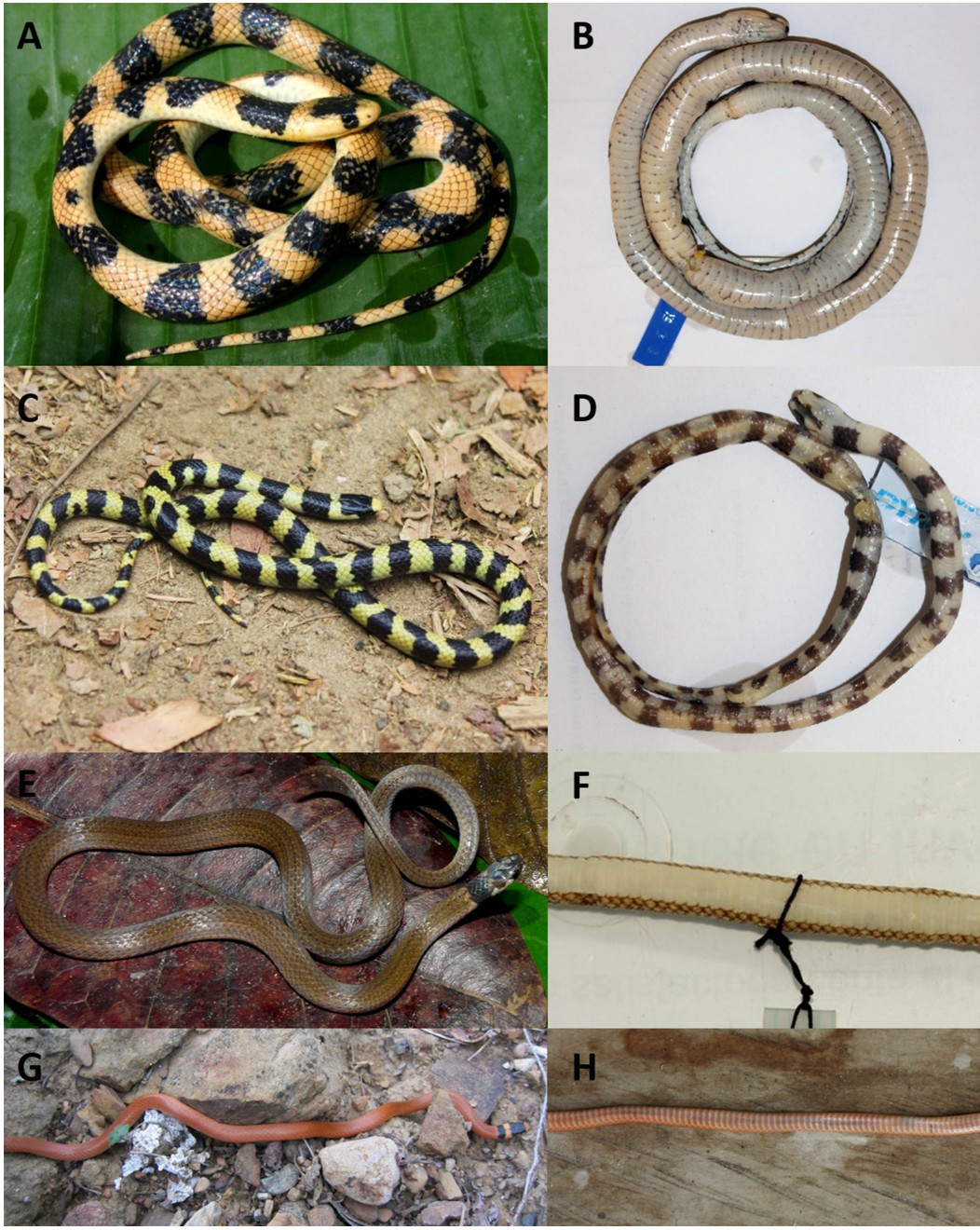

**Figure 5** Dorsal (left) and ventral (right) views of the species of *Tantilla* fom Peru: *T. tjiasmantoi* sp. nov. female holotype CORBIDI 7726 (A, B) and male paratype ZFMK 95238 (C, D); *T. melanocephala* (E, F) from Bahuaja-Sonene, Madre de Díos (photographs by Roy Santa Cruz); *T. capistrata* (G, H) from near Santa Catalina de Chongoyape, Lambayeque.

### Cranial osteology

The snout is terminated by a small, single, toothless, median premaxilla. The triangular-shaped dorsal process of the premaxilla is not contacting the anterior portion of the two nasals. The paired nasals are slightly convex and oval-shaped in dorsal view, medially in contact and separate anteriorly to from an interspace. Their posterior edges are in contact with the anterior inner edges of the frontals. The paired frontals are dorsally flat, subrectangular, longer than broad, slightly larger than the nasals, and in close medial contact. The posterolateral edge of each frontal forms a supraorbital ridge which is perforated with two supraorbital foramina. The lateral surface of the frontal forms a major portion of the mesial wall of the orbit. The prefrontal is loosely attached to the anterolateral edge of the frontal. In lateral view, the prefrontal is anteriorly contacting the posterior edge of the nasal and the septomaxilla, and the ventral border is contacting the dorsal surface of the maxilla. The frontals articulate posteriorly with the fused parietals. In dorsal view, the parietal is flat, oval-shaped except for the more or less straight anterior border along the suture with the frontals. It is the largest of the cranial elements and like the frontals, extends laterally far down, reaching ventrally to the basisphenoid. The parietal forms the posterior portion of each orbit. Its posterolateral borders suture with the prootics and its posterior border sutures with the supraoccipital. The postorbitals are narrow, elongate, flattened, slightly curved bones, larger than the prefrontals. They articulate with the anterolateral surface of the parietal and form the dorsoposterior boundary of each orbit. The single supraoccipital is flat, subpentagonal, and broader than long. Laterally it unites with the prootics and posteriorly with the exoccipitals. The paired prootics are largely separated by the broad supraoccipital. Each prootic is subtrapezoidal in dorsal view and is posteriorly bordered by the exoccipital. Laterally it extends far down, reaching ventrally the basisphenoid and the basioccipital. Each prootic has two large foramen of which the anterior one is anteriorly not enclosed by bone. The posterior foramen is larger than the anterior one. Other smaller foramina pierce each prootic. Posterodorsally each prootic is fused with the elongate, spine-like supratemporal, which connects the posterolateral dorsal part of the skull with the proximal end of the quadrate. Both supratemprals are almost parallel to each other, converging only slightly posteriard. The exoccipitals are dorsally flat, form the posterolateral walls of the braincase and are joined together by a mid-dorsal suture. Ventrally they are resting upon the basioccipital. The basioccipital is pentagonal-shaped and joined anteroventrally with the basisphenoid along a straight transverse suture. The basisphenoid is applied anteriorly without a suture to the narrow, elongate parasphenoid, forming a single bone, which extends anteriorly into the rostrum area and becomes the floor of the orbit. Each maxilla is a curved bar with a small horizontal dorsoposteriorly-pointing process, about midway on the mesial border, that articulates with the ventral surface of the prefrontal. Anterodorsally articulates with the posteroventral surface of the septomaxilla. The posterior end of the maxilla is broadened and received by the flattened, pincers-shaped ectopterygoid, which connects it to the pterygoid. Each maxilla bears sockets for about 18 prediastemal, slightly recurved teeth, followed by two slightly enlarged, ungrooved fangs. The ectopterygoid do not bear teeth. The narrow palatine bears 13 or 14 teeth. The posterior end of each palatine articulates with the anterior end of the pterygoid. At

about the height of the third and fourth tooth each palatine possesses a broad, horizontally flattened, lateral maxillary process. At about the height of the sixth to eighth tooth each palatine possesses a similar but median choanal process with the apex directed anteriorly. The pterygoid is a slightly curved, flattened bar with about 17 teeth along the medial border. The subtriangular quadrate articulates with the lateral border of the supratemporal and its distal surface articulates with the mandibular condyle. The mandible is composed of two jaw bones. The anterolateral portion of the jaw is formed by the dentary, which contains a row of about 22, slightly recurved teeth. The longer proximal part of each jaw is without teeth (Fig. 6).

### Hemipenial morphology

The hemipenes are unilobed, unicalyculate and noncapitate with a single sulcus spermaticus. The apical part is mostly uniformly spinulate. The hemipenial body is more or less uniformly ornamented with long and thin spines, which increase in size towards the base (Fig. 7).

### Etymology

The species is dedicated to Wewin Tjiasmanto (Indonesia) in recognition of his support of nature conservation and taxonomic research through the BIOPAT initiative.

### Distribution and natural history

This species is so far known from the southern portion of the seasonally dry forest along the Marañón River and its tributaries, from near Santa Rosa de Marcamachay at the Río Crisnejas, Province Cajabamba, and from near Laguna de Pías, Province Pataz, both Department of La Libertad, at elevations of 1,154 m and 1,726 m a.s.l., respectively (Figs. 8 & 9). The female CORBIDI 7726 was detected on 7th of January 2010 at 12.30 pm resting on a stone. The male ZFMK 95238 was detected on 12th of October 2010 at 8.15 pm on pebbly-clayey ground. Air temperature when animals were sighted was 33.3 °C and 28.1 °C, respectively.

## DISCUSSION

Despite the comparatively close localities (>70 km air distance) and similarities in scale counts and arrangement of scales, the conspicuous differences of both specimens in body size and ventral coloration created some doubt if they represent the same species. A pairwise analysis of 1,576 bp derived from the mitochondrial gene 16S and the nuclear gene RAG1 showed only a single difference in one base position in the RAG1 fragment, thus strongly supporting the assumption that both specimens belong to the same species. The collection and examination of further specimens is needed to determine whether the differences in size and color pattern are a product of sexual dimorphism or are referred to a different cause (e.g., age dependence, geographic variation).

In order to get a general idea of the phylogenetic position of the new species described herein, we performed phylogenetic analyses based on 12S rRNA and 16S rRNA. We did not attempt to conduct a taxonomically extensive analysis of South American Colubridae, instead we preferred to include only those species, for which the two gene regions sequenced

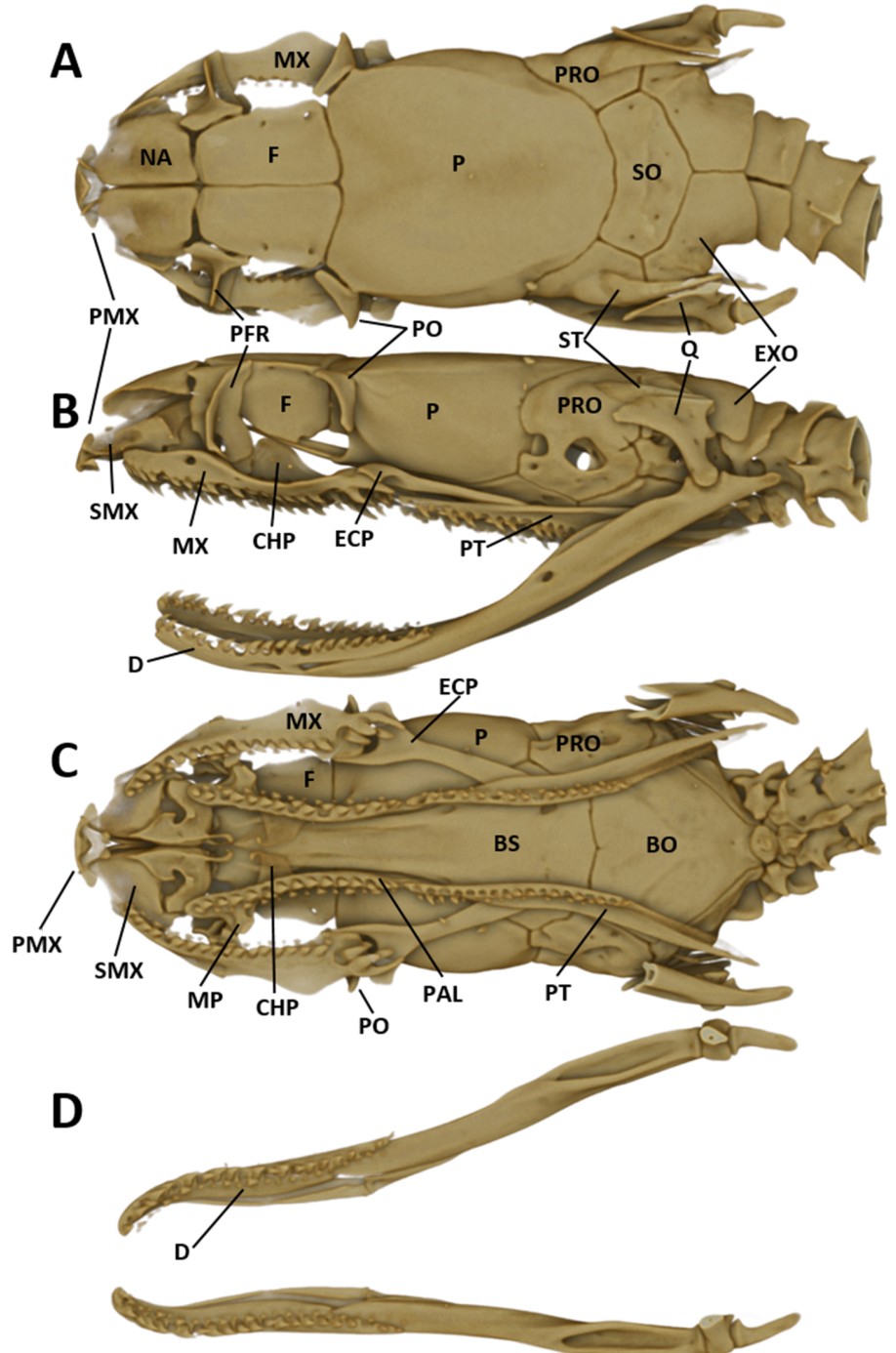

**Figure 6  Micro-CT images of the male paratype of *Tantilla tjiasmantoi* sp. nov. from La Libertad: dorsal (A), lateral (B), and ventral views of the skull (C, lower jaw removed), and dorsal view of lower jaw (D).** See 'Materials and Methods' section for explanation of the abbreviations of the skull structures.

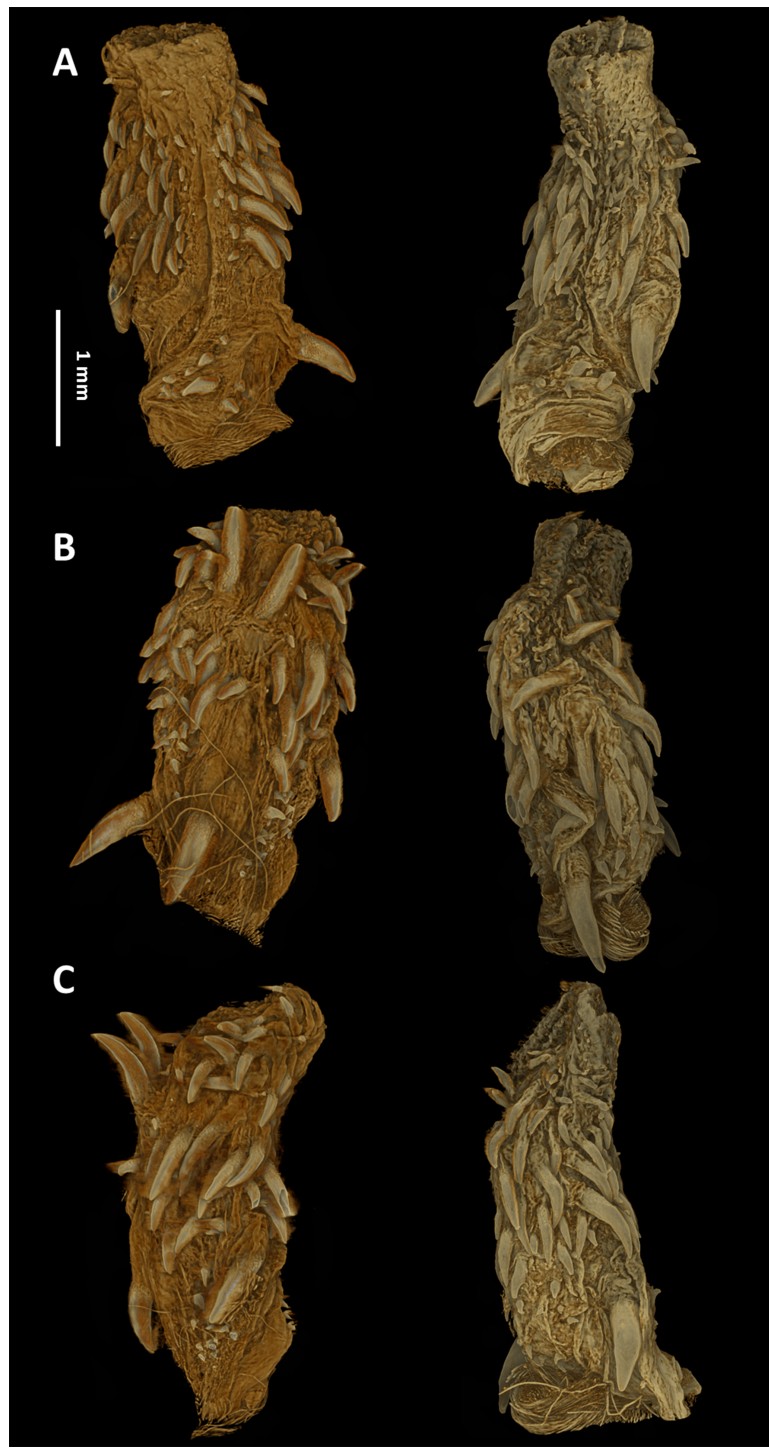

**Figure 7** Micro-CT images of the hemipenes of the male paratype of *Tantilla tjiasmantoi* sp. nov. from La Libertad: sulcate (A), asulcate (B), and lateral views (C) of the left (left) and right (right) hemipenis. Left hemipenis was scanned dry, whereas right hemipenis was scanned in alcohol.

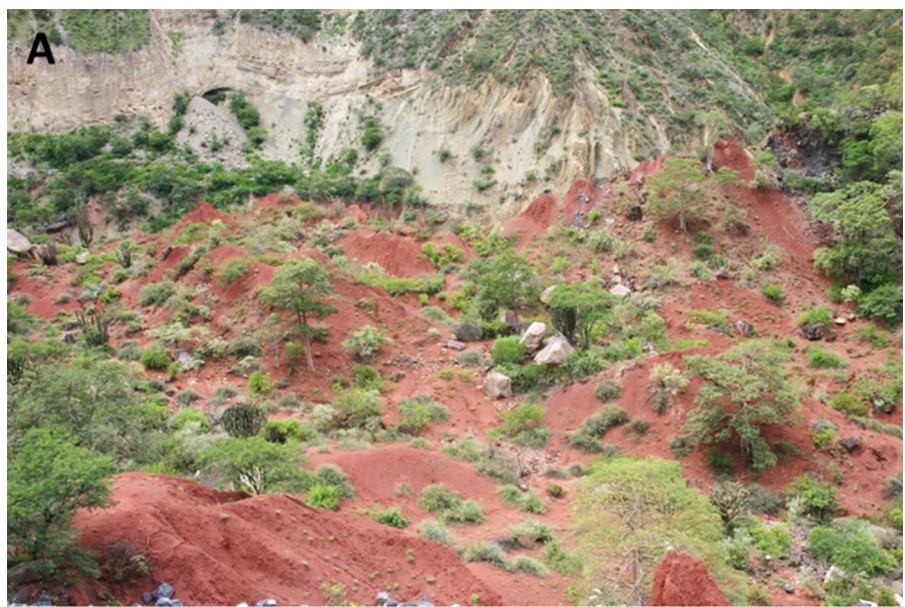

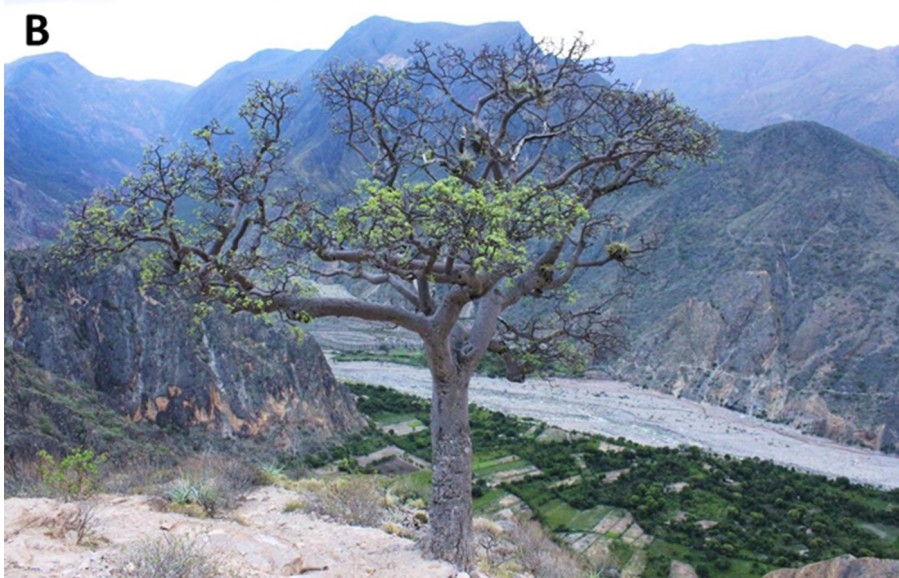

**Figure 8** **Habitat and localities of the holotype of *Tantilla tjiasmantoi* sp. nov. CORBIDI 7726 near Laguna de Pías, La Libertad, Peru (A), and male paratype ZFMK 95238 near Santa Rosa de Marcamachay, La Libertad, Peru (B).**

in this study were available in GenBank. Our phylogenetic tree based on mitochondrial DNA (Fig. 2) corroborates the assignment of our new species into the genus *Tantilla*. Both the monophyly of the sampled *Tantilla* and the conspecificity of our two dimorphic specimens are well supported by the analyses. However, intrageneric relationships remain dubious due to the few species with genetic data available. Moreover, scutellation characteristics and the comparison of the skull morphology via micro-CT scans (Fig. 1) strongly support this hypothesis.

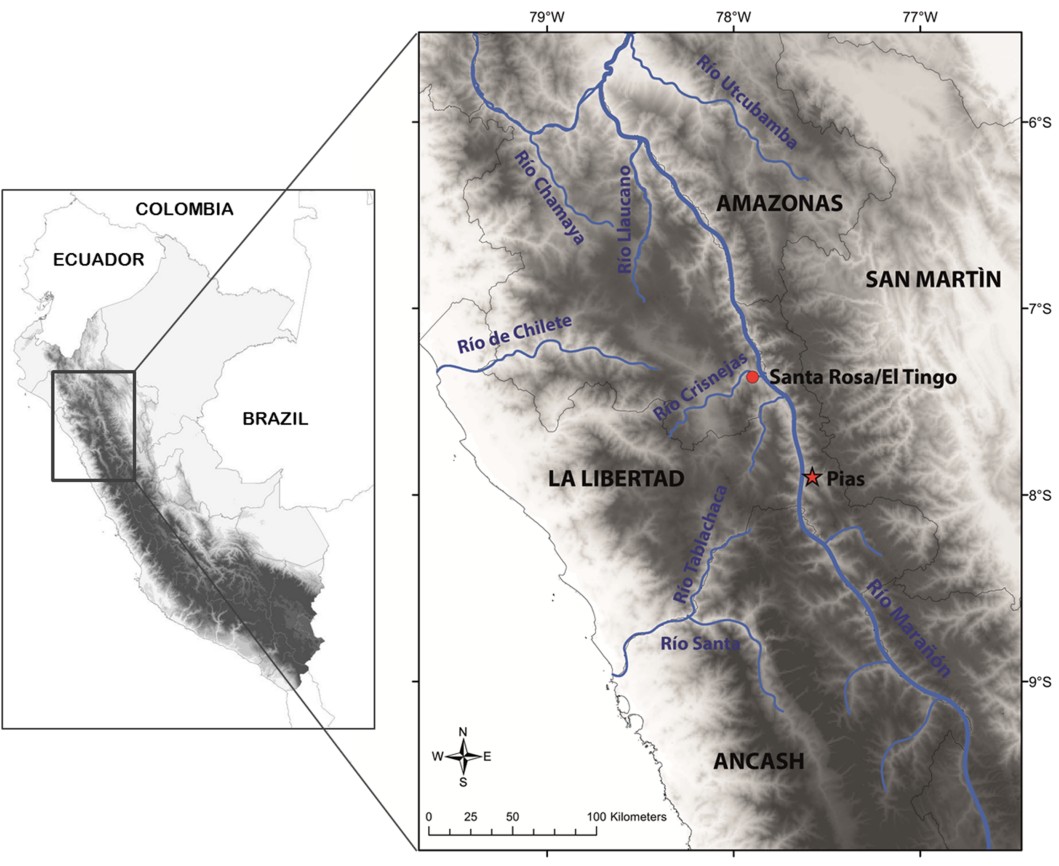

**Figure 9** Map with type locality of *Tantilla tjiasmantoi* sp. nov. (red star), and locality of male paratype (red dot) in the northern Peruvian Andes.

With currently 62 species assigned to this genus it represents the second largest genus of New World colubrid snakes, after the genus *Atractus*. However, taxonomic approaches are limited by the fact that pretty much nothing concrete is known about the phylogenetic relationships within the genus *Tantilla*, and its monophyly has not yet been tested adequately. Only few specimens representing fewer than ten species of *Tantilla* have been included in previous DNA-based studies (e.g., *Vidal et al., 2000*; *Lawson et al., 2005*; *Burbrink & Myers, 2015*; *Schrey et al., 2015*; *Chambers & Hebert, 2016*). Thus, *Tantilla* could benefit from a thorough taxonomic treatment involving a stronger genomic sampling component. Such a revision would test species concepts, update their known distributions, reveal their genetic diversity and give some clues about their evolutionary history. Furthermore, with robust molecular data generic designation could be proposed with more confidence.

The habitat of this new species is gravely threatened due to human interventions, such as deforestation, mining activities, and intended dam constructions for hydroelectric projects. To date no protected area has been established in the Marañón river valley. We hope that this beautiful and untypically colored new *Tantilla* could serve as a flagship species, together with several other endemic species of reptiles and birds, for the establishment

of conservation strategies in this region. Unless these strategies are implemented the biodiversity found in this unique habitat, including the new, endemic species described here, may fall into serious decline.

## ACKNOWLEDGEMENTS

Peter Rühr and Morris Flecks (both ZFMK) kindly helped with the micro-CT imaging and phylogenetic analysis, respectively. We are indebted to Hussam Zaher for sharing his expertise on hemipenes morphology with us. Roy Santa Cruz kindly provided photographs of a live specimen of *T. melanocephala*. The Ministerio de Agricultura, Peru, kindly provided collecting and export permits. We thank Alfredo Beraún, Erick Hoyos Granda, and Manuel Palacios Panta for assistance during the fieldwork. We are indebted to the residents of Santa Rosa and Laguna de Pías for their hospitality, support and assistance during the fieldwork.

## APPENDIX: ADDITIONAL SPECIMENS EXAMINED

*Tantilla capistrata*: PERU: Lambayeque: Chaparrí, 446 m a.s.l. (ZFMK 85028).
*Tantilla melanocephala*: BOLIVIA: Santa Cruz: Florida, Pampagrande 446 m a.s.l. (ZFMK 75041).
*Tantilla relicta*: USA: Florida: Bushnell (ZFMK 84387).

### Funding

Claudia Koch received a travel grant from the Deutscher Akademischer Austauschdienst (DAAD), and financial support from the Alexander Koenig Stiftung (AKS) and the Alexander Koenig Gesellschaft (AKG). The funders had no role in study design, data collection and analysis, decision to publish, or preparation of the manuscript.

### Grant Disclosures

The following grant information was disclosed by the authors:
Deutscher Akademischer Austauschdienst (DAAD).
Alexander Koenig Stiftung (AKS).
Alexander Koenig Gesellschaft (AKG).

### Competing Interests

The authors declare there are no competing interests.

### Author Contributions

- Claudia Koch conceived and designed the experiments, performed the experiments, analyzed the data, contributed reagents/materials/analysis tools, wrote the paper, prepared figures and/or tables, reviewed drafts of the paper.
- Pablo J. Venegas conceived and designed the experiments, reviewed drafts of the paper.

## Animal Ethics

The following information was supplied relating to ethical approvals (i.e., approving body and any reference numbers):

The Ministerio de Agricultura, Peru granted research (0020-2009-AG-DGFFS-DGEFFS, 0424-2010-AG-DGFFS-DGEFFS) and export permits (003983-AG-DGFFS).

## DNA Deposition

The following information was supplied regarding the deposition of DNA sequences:

The RAG1, 12S, and 16S sequences for the new species described here are accessible via GenBank accession numbers KY006873, KY006874, KY006875, KY006876, KY006877.

## Data Availability

The raw data has been supplied as a Supplemental File.

## New Species Registration

The following information was supplied regarding the registration of a newly described species:

*Tantilla tjiasmantoi*

taxon LSID: urn:lsid:zoobank.org:act:15B70DA0-55D5-47D9-8383-453869F3B530.

publication LSID: urn:lsid:zoobank.org:pub:00EBF842-3AFA-4913-B381 -95BDF86DAFAB.

## Supplemental Information

Supplemental information for this article can be found online at http://dx.doi.org/10.7717/peerj.2767#supplemental-information.

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
