# Peer review of "A large and unusually colored new snake species of the genus Tantilla (Squamata; Colubridae) from the Peruvian Andes"

_PeerJ, doi:10.7717/peerj.2767_

## Round 0.1 · original submission · Major Revisions

Dear authors

As you can see some of of reviewers were not very satisfied with your present ms. If you are willing to revise the ms according to the suggestions of the reviewers, we will review it another time.

Greetings
Michael Wink
Academic editor

·

Basic reporting

This study describes a new species of Tantilla for the Peruvian Andes. The information provided by the authors is really important, because besides the ‎importance of the studied area, few studies have been conducted about the herpetofauna of this locality. ‎
Overall, the authors’ style is clear and the English language is suitable for the journal, however a minor English review is recommended to correct typos and ‎grammar mistakes. ‎
Information about the systematic implication within Tantilla genus can be added in the introduction.

Experimental design

Methods and analysis are robust, though I suggest including a better description of the area and the authors might include a map with a schematic range of ‎the recognized species of Tantilla for Peruvian Andes and the new species locality. ‎
I missed some statistical analysis to verify if the morphological biometric variation among male and female specimens are significant different.‎
I would like also to see a statistical test to verify the morphological variation among the three different Tantilla’s species from Peru. ‎

Validity of the findings

The manuscript is original and brings new information to the scientific community. It is valuable and interesting for the conservation of the area and snake species.

Additional comments

After few corrections the manuscript can be accepted to the journal. The overall quality and contents are good.

Reviewer 2 ·

Basic reporting

The aim of this manuscript is the description of a new species of the snake genus Tantilla by the use of different sources of evidence, such as morphology, morphometrics and molecular data. However, the authors do not clearly provide evidence for the existence of a new taxon because very few species of the genus were analyzed for comparison. The examined specimens may very well belong to a new species, though. But the differences should be more explicitly presented by the use of a more robust database than the one presented by the authors. Of the 61 species of the genus only 4 were included in phylogenetic analyses – only ONE of the two species that occur in Peru was analyzed. In the morphometric analysis only three specimens were analyzed by X-ray analysis and again only one of the two species from Peru. For the description of a new species it is important to show evidence that the specimen collected does not belong to any previously described species. The necessary evidence is not adequately presented by the authors.

Experimental design

Although different sources of evidence were used for the description of the new species, few species of the genus Tantilla were analyzed. The minimum expected for this description would be a comparative analysis among the taxon novum and the other 12 species having distribution in South America. See comments in the reviewer material.

Validity of the findings

There seems to be distinct morphology, i.e., size and differences of color of scales of the Taxon novum. The genetic analysis and tridimensional morphometry are insufficient; a robust comparison to other species of the genus is necessary.

Additional comments

I suggest that all 12 species of the genus Tantilla occurring in South America be sequenced and analyzed by X-ray.

Annotated reviews are not available for download in order to protect the identity of reviewers who chose to remain anonymous.

·

Basic reporting

There are a few minor language problems - see my comments in the attached pdf.

Experimental design

n.a.

Validity of the findings

good

Additional comments

Nice article on a beautiful species. Congratulations! I have a few comments in the attached pdf - all minor issues.

---

## Round 0.2 · accepted · Accept

Dear authors

Thank you for your revision. Now your MS can be accepted.

Congratulations!

Greetings
Michael Wink
Academic editor

Reviewer 2 ·

Basic reporting

The authors have now dealt with all my comments on the first round of revision and I feel the manuscript is greatly improved

Experimental design

The revised manuscript clarifies many of the issues in the earlier version, so it is much stronger now

Validity of the findings

Distinct morphology of the Taxon novum were adequately presented in this version of the manuscript